# Attention-deficit/hyperactivity disorder and serial missed appointments in general practice

**Ross McQueenie[1], David A. Ellis[2]\*, Andrea Williamson[1], Philip Wilson[3,4]**

**1** General Practice and Primary Care, School of Health and Wellbeing, MVLS, University of Glasgow, Scotland, **2** School of Management, University of Bath, Bath, United Kingdom, **3** Section of General Practice, Department of Public Health, University of Copenhagen, Copenhagen, Denmark, **4** Centre for Rural Health, Institute of Applied Health Sciences, University of Aberdeen, United Kingdom

\* dae30@bath.ac.uk

**Data Availability Statement:** These data were made available from NHS Scotland but restrictions apply to sharing. Specifically, data were used under license for the current study, and so are not publicly available. Requests to access these data in

## Abstract

Missingness' in health care has recently been flagged as a major challenge due to associations between missing multiple appointments and poor long-term outcomes. Patients with a range of mental health diagnoses, for example, can have difficulties maintaining continuity of care and miss appointments with adverse consequences, including increased mortality. This is especially problematic for those with a diagnosis of ADHD because it is relatively common, with an estimated prevalence of 2–5% of the population. This study therefore aimed to a) establish the prevalence of recorded ADHD b) characterise and compare individuals with and without ADHD in terms of health and social variables, and c) assess whether ADHD was associated with an increased risk of missing scheduled appointments in general practice. Using administrative data from 136 Scottish general practices, patients with at least one GP appointment between September 2013–2016 we identified those with ADHD based on diagnoses and prescribing data. Each case was matched (sex and age) to five randomly selected GP attendees. Groups were compared regarding health, social status and missed appointments. All results were stratified by age <18 or ≥18 years. Among 824,374 GP patients we identified 2,452 with a record of ADHD (0.8% among those <18 years; 0.2% age ≥18 years). ADHD was associated with living in socially deprived areas of Scotland, and multimorbidity was more frequent in adults (p<0.01). Adjusting for the number of total appointments made, ADHD was associated missing GP appointments (<18 years: OR = 1.6, 95%CI = 1.4–1.9; ≥18 years: OR = 1.9, 95%CI = 1.7–2.2). Annually 21% in those <18 and 38% of those age ≥18 years missed at least one GP appointment. The prevalence of recorded ADHD in Scottish general practice is low but comparable to other studies using data derived from medical records. More importantly, this is the first study to demonstrate an association between ADHD and missingness in general practice. The findings have important implications for health services concerning the early diagnosis and treatment of ADHD.

the same manner as the authors can be made to http://www.escro.co.uk/ for general practice data and to https://www.isdscotland.org/Products-and-Services/eDRIS/ who will host the analysis in a secure environment.

**Funding:** This study was funded by a grant awarded to AW, DAE and PW from the Chief Scientist Office, Scottish Government (reference CZH/4/1118), with Safe Haven and data linkage costs supported in lieu by the DSLS at Scottish Government. The funding sources for this analysis had no influence over study design, data collection, data analysis, data interpretation, the writing of the report or the decision to submit for publication. https://www.cso.scot.nhs.uk/.

**Competing interests:** The authors have declared that no competing interests exist.

## Introduction

General practitioners (GPs) have a vital role in many health care systems. They not only treat a range of mental and physical health conditions, but also serve as gatekeepers to the wider health care system and provide continuity of care. However, some patients with, for example mental health problems may have difficulties maintaining continuity of care with their GP and miss appointments with potential adverse consequences [1]. 'Missingness' in health care has recently been further evaluated; in patients with long-term physical health problems missing two or more appointments annually was associated with a 2–3-fold increased mortality risk within a 16 months follow-up period. In patients with long-term mental health conditions the corresponding finding was a 7–8 fold increase in mortality risk, often attributable to non-natural causes [2]. Thus, missingness is not just inconvenient for health care planners but may be an important indicator of patient deterioration or may represent missed opportunities for delivering needed health care to the patient.

A number of psychiatric conditions could account for some of the association between missing appointments and premature mortality in patients with mental health problems, including psychosis [3], personality disorders [4], depression [5] and attention-deficit/hyperactivity disorder (ADHD) [6]. ADHD is of particular interest because it is relatively common, with an estimated prevalence of 2–5% of the population [7, 8] but its prevalence in health service records is lower than the population prevalence in many countries [9, 10]. While ADHD is not always considered to be a mental health problem by the ADHD community, people with an ADHD diagnosis will likely have an above average need to access GPs irrespective of age. For example, ADHD is associated with various mental (e.g. depression, anxiety, and substance use) [11–13] and physical (e.g. physical injuries) [14] health conditions across the lifespan that require follow-up and treatment.

Among adults with ADHD surveyed in high-income countries, fewer than one third had received any professional help during the previous 12 months, and only 5% had received treatment specifically for ADHD during the same period [8]. Many factors could contribute to this low level of therapeutic support including stigma and lack of awareness or knowledge among health care providers about the importance and benefits of treatment [15]. Furthermore, patients with ADHD may struggle to access services due to the cognitive impairments associated with the condition (e.g. planning, initiating, and memory problems), which could impact on their ability to schedule visits as well as to remember appointments that have been made. Missed appointments may be of clinical significance as an early sign of deteriorating physical or mental health, and, more importantly, may lead to adverse health outcomes [2].

A few studies have examined missed appointments in specialist services in relation to ADHD [16–18]. There are, however, no published studies examining associations between patterns of missed appointments and ADHD in general practice. Since the population attending general practice is unselected and access is near-universal in the UK, GP service use data provides a unique opportunity to further understand or potentially replicate previous findings from secondary care. While general practice appointment data are difficult to access at national level, this has recently become possible in Scotland [2, 19]. Here we present data on GP appointment attendance among patients with a recorded diagnosis of ADHD or a history of prescribed psychostimulant medication.

### Aims

We aimed to a) study how many children and adolescents (<18 years of age) and adults (≥18 years) are recorded in a large Scottish GP dataset with a diagnosis of ADHD or a prescription for ADHD medication, b) characterise and compare individuals identified with and without

ADHD in terms of physical and mental health problems, and c) assess patterns of missed appointments by comparing age/sex matched patients with and without ADHD.

## Materials and methods

### Ethics statement

Letters of comfort were issued by the West of Scotland NHS Ethics Committee and the University of Glasgow College of Medical, Veterinary & Life Sciences Ethics Committee confirming that the full study did not need NHS ethics approval as this used routine data that had been extracted in an anonymised format. Public Benefit and Privacy Panel approval for use of the data was granted by NHS Information Services Scotland in December 2016.

This is a retrospective cohort study based on a sample of nationwide data from National Health Service Scotland (NHS) general practices. The exposure in the study was ADHD, and a randomly selected cohort of individuals without ADHD was matched on sex and age in a 5:1 ratio. Matching was done on sex and age because studies on ADHD using administrative data consistently find an overrepresentation of males and younger people in ADHD samples [9]. A 5:1 matching ratio was chosen to increase power of comparisons between those with and without an ADHD diagnosis. The outcome of interest was missed appointments in general practice.

Within a nationally representative sample of 136 general practices, 824,374 patients who had scheduled at least one appointment during a 3-year period from September 2013 to 2016 were identified. Data were extracted by a Trusted Third Party for the NHS, anonymised and linked to a unique patient identifier in the Scottish NHS Safe haven for analysis.

Data were cleaned to ensure that each appointment was logged as attended or missed (did not attend). This was primarily based on the "in" and "out" time recorded for that appointment. If this was recorded as "0" then the appointment was classified as did not attend. It was not possible to code for appointments that were cancelled or where a patient "Was not brough'.

Appointments regarded as non-face-to-face consultations were removed. These were defined in our dataset as "Administrator", "Receptionist", "Secretary", "Other Admin and Clerical", "Practice Manager", and "Unknown". Any patients with a blank registration date were also removed along with those who were not registered as patients with the practice in the study period. Finally, appointments for which the waiting time was less than 0 (negative) and those that lasted less than 2 min were removed. We derived the 2 min distinction from a pilot study [20].

Diagnostic status was defined using all data held in the GP record, dating back to birth in most cases since records are transferred when a patient moves between practices in the UK. A diagnosis of (ADHD) was defined in two ways; first from Read codes that GPs had used to code a diagnosis of ADHD in the patient's clinical record (usually derived from mental health service letters) at any time up to 2016. The codes were checked against a previously published UK GP database study, and five additional codes were included [21]. Second, patients who were prescribed any ADHD medication up to 2016 were also included. British National Formulary (BNF) ADHD drugs from the BNF 2012 and BNF 2018–19 were reviewed, and generic and brand names of drugs were collated. This was compared with the medication list from Newlove-Delgado et al. [21]. There is some overlap between medications used in ADHD and narcolepsy so patients with the Read codes relating to narcolepsy were excluded.

In order to characterise the sample, and adjust for potential comorbidity, information was retained in the ADHD and non-ADHD patient data regarding history of any recorded psychiatric or physical morbidity using Read codes and prescribing data as described elsewhere [2,

22]. Factors known to be associated with ADHD which might confound a potential association between ADHD and missing GP appointments were also extracted from the GP dataset: problematic alcohol use, problematic psychoactive substance use, learning difficulties, as well as socio-economic status.

## Analysis

Patients were categorised based on average rates of non-attendance (missed appointments) over the three year study period from September 5, 2013, until September 5, 2016, as follows: zero missed appointments (zero group); low number of missed appointments, $< 1$ per year (low group); medium number of missed appointments, 1–2 per year (medium group); and high number of missed appointments, $> 2$ per year (high group). Number of long-term conditions (LTCs) were calculated for each participant based on 43 individual prevalent LTCs as described by Barnet et al. [22]. LTC categories were created to measure number of physical LTCs, number of mental LTCs and number of both physical and mental LTCs.

All analyses were carried out on age and sex matched case-control cohorts as described above and results were stratified for those $< 18$ years of age and $\geq 18$ years of age based on the individual's age at the time of data-extraction. Data were then analysed descriptively and compared based on ADHD status. Tests were considered significant if $p < 0.01$ with values derived from $\chi^2$ tests. To ascertain whether there was any increased risk of missing appointments among ADHD patients compared to case-controls, we then used negative binomial modelling with number of missed appointments as an outcome variable. These models were offset for number of appointments made in order to account for the possibility that either those with or without ADHD had made more GP appointments.

The data were processed and analysed using R version 3.6.1.

## Ethics

The data contained within this study did not require ethical approval since it was defined as a service evaluation. We obtained a letter of comfort from the West of Scotland NHS Ethics Committee and the University of Glasgow College of Medical, Veterinary & Life Sciences Ethics Committee confirming that the full study did not need health service ethics permissions.

## Results

### Prevalence

Using the definitions outlined previously, we identified 1,148 young people and 1,304 adults with ADHD. Among young people 1,038 (90.4%) had ADHD prescriptions, and 439 (38.2%) an ADHD Read code. Among the adults (age $\geq 18$ years) the corresponding numbers were 1,126 (86.3%) and 201 (15.4%). The overall recorded prevalence of ADHD in the total GP dataset was 0.3%. For those $< 18$ years, the recorded prevalence was 0.8% and for adults (age $\geq 18$ years) the recorded prevalence was 0.2%. We identified that 84% of all recorded ADHD cases were observed in patients under 35 years of age, and that the majority of ADHD patients were male (n = 1,828, 74.6%).

### Sample characteristics

The ADHD sample was matched on age and sex to 5,740 patients under 18 years and 6,520 adult controls respectively. Tables 1 and 2 present comparisons of those with and without recorded ADHD stratified on age.

**Table 1. Characteristics of children and adolescents (age 0–17 years) with and without recorded ADHD diagnosis/prescriptions.**

| | | ADHD (n = 1,148) | | Controls (n = 5,740) | | $\chi^2$ | p-value |
|---|---|---|---|---|---|---|---|
| | | N | % | N | % | | |
| Age-group | 0–5 years | 5 | 0.4 | 25 | 0.4 | 0.0 | 1.00 |
| | 6–10 years | 389 | 33.9 | 1946 | 33.9 | | |
| | 11–14 years | 438 | 38.2 | 2188 | 38.1 | | |
| | 15–17 years | 316 | 27.5 | 1581 | 27.5 | | |
| Sex | Males | 934 | 81.4 | 4670 | 81.4 | 0.0 | 1.00 |
| | Females | 214 | 18.6 | 1070 | 18.6 | | |
| Physical multi-morbidity | None | 869 | 75.7 | 3324 | 57.9 | 127.1 | <0.01 |
| | One to three | 279 | 24.3 | 2415 | 42.1 | | |
| | Four+ | - | - | - | 0% | | |
| Mental multi-morbidity | None | 1086 | 94.8 | 5459 | 95.1 | 0.0 | 0.62 |
| | One to three | 62 | 5.2 | 281 | 4.9 | | |
| | Four+ | - | - | - | - | | |
| Combined physical and mental morbidity | No | 1128 | 98.3 | 5624 | 98.0 | 0.3 | 0.38 |
| | Yes | 20 | 1.7 | 116 | 2.0 | | |
| Social deprivation index (SIMD) quintile | 1 (most deprived) | 399 | 34.8 | 1511 | 26.3 | 86.1 | <0.01 |
| | 2 | 141 | 21.0 | 1099 | 19.1 | | |
| | 3 | 196 | 19.6. | 951 | 16.5 | | |
| | 4 | 170 | 14.8 | 1079 | 18.9 | | |
| | 5 (most affluent) | 117 | 10.1 | 977 | 17.1 | | |
| | Missing | 25 | 2.2 | 122 | 2.1 | | |

SIMD: Scottish Index of Multiple Deprivation, reported in quintiles

There was a markedly higher recorded prevalence of ADHD among younger patients. Patients with ADHD were more likely to reside in more socio-economically deprived areas.

As the prevalence of specific comorbidities was low in the child and adolescent sample, these data are not reported here. In the adult sample, the recorded prevalence of problem alcohol use and recorded non-prescribed psychoactive drug use was almost double that of control patients. Learning difficulty-related diagnoses were also substantially more prevalent in the ADHD sample than among the control group.

## Missed appointments

Patients with ADHD were substantially more likely to miss multiple GP appointments during the study period and this was observed for both those <18 years and ≥18 years (p<0.01) (Table 3). Adjusting for the between group differences related to number of appointments made, we found an increased risk for adults for missing appointments (OR = 1.9, 95% CI = 1.7–2.2) and for those age <18 years (OR = 1.6, 95%CI = 1.4–1.9). It can be seen that adults are more likely to miss GP appointments than young people, but repeated missing of appointments occurs across the lifespan and confidence intervals are overlapping.

## Discussion

This is the first study to estimate the extent to which patients with ADHD are more likely to miss appointments in general practice than patients without ADHD. The prevalence of recorded ADHD diagnoses in our sample of over 800,000 Scottish patients was 0.3%, with 84% of all recorded cases in patients under 35 years of age. Adult patients with recorded ADHD

**Table 2. Characteristics of adults (age 18–74 years) with and without recorded ADHD diagnosis/prescriptions.**

| | | ADHD (n = 1,304) | | Controls (n = 6,520) | | $\chi^2$ | p-value |
|---|---|---|---|---|---|---|---|
| | | N | % | N | % | | |
| Age-group | 18–30 | 847 | 65.0 | 4234 | 65.0 | 0.0 | 1.00 |
| | 31–45 | 245 | 18.8 | 1225 | 18.8 | | |
| | 46–60 | 150 | 11.5 | 751 | 11.5 | | |
| | 61–75 | 51 | 3.9 | 255 | 3.9 | | |
| | 76–90 | 10 | 0.8 | 50 | 0.8 | | |
| | Over 90 | 1 | 0.1 | 5 | 0.1 | | |
| Sex | Males | 894 | 68.6 | 4470 | 68.6 | 0.0 | 1.00 |
| | Females | 410 | 31.4 | 2050 | 31.4 | | |
| Physical multi-morbidity | None | 675 | 51.8 | 3198 | 49.0 | 9.3 | 0.01 |
| | One to three | 580 | 44.5 | 3147 | 48.3 | | |
| | Four+ | 49 | 3.7 | 175 | 2.7 | | |
| Mental multi-morbidity | None | 701 | 53.8 | 4089 | 62.7 | 49.2 | <0.01 |
| | One to three | 585 | 44.9 | 2403 | 36.9 | | |
| | Four+ | 18 | 1.4 | 28 | 0.4 | | |
| Combined physical and mental morbidity | No | 948 | 72.7 | 5219 | 80.0 | 36.7 | <0.01 |
| | Yes | 356 | 27.3 | 1301 | 20.0 | | |
| Substance abuse | Yes | 102 | 7.8 | 272 | 4.2 | 31.0 | <0.01 |
| Alcohol abuse | Yes | 69 | 5.3 | 189 | 2.9 | 18.8 | <0.01 |
| Learning disability | Yes | 59 | 4.5 | 56 | 0.9 | 98.3 | <0.01 |
| Social deprivation index (SIMD) quintile | 1 (most deprived) | 335 | 26.3 | 1615 | 24.7 | 30.1 | <0.01 |
| | 2 | 186 | 22.5 | 1283 | 19.7 | | |
| | 3 | 254 | 19.9 | 1152 | 17.6 | | |
| | 4 | 211 | 16.6 | 1175 | 18.0 | | |
| | 5 (most affluent) | 187 | 14.7 | 1158 | 17.8 | | |
| | Missing | 31 | 2.4 | 137 | 2.1 | | |

SIMD: Scottish Index of Multiple Deprivation, reported in quintiles

diagnoses had higher levels of mental and combined physical/mental co-morbidity than an age- and sex-matched control sample. This was not replicated in the child and adolescent sample. Among the adults, these comorbidities included learning difficulties as well as problem alcohol and drug use, all of which could contribute to difficulties attending GP appointments.

The low prevalence of recorded ADHD (0.3%) reflects findings from other contexts [10]. Given the impairing and life-long nature of ADHD, we might expect that a substantial proportion of people with ADHD in the general population would have received an assessment for ADHD diagnosis and would continue to attend services and receive treatment. Nevertheless, the incidence and prevalence of diagnosed and treated ADHD in administrative datasets from many countries is lower than expected [9, 10, 23, 24]. A recent study reported on the annual prevalence of prescriptions for ADHD medication in various countries including data from the US, Canada, Australia, UK, Scandinavia, Central and Southern Europe, Taiwan, Japan, and Hong-Kong [10]. The study found large between-country variations, but most countries had an annual prevalence of treated ADHD in children age 3–18 years of 1–2% and the pooled annual prevalence in adults was 0.39% [10]. Thus, despite increases in the administrative prevalence, there still seems to be, at least in some countries, a relatively large proportion of

**Table 3. Scheduled and missed appointments among ADHD cases and controls.**

| Children and adolescents | ADHD (n = 1,148) | | Controls (n = 5,740) | | t/ $\chi^2$ | p-value |
|---|---|---|---|---|---|---|
| | M/N | SD/% | M/N | SD/% | | |
| Mean number of appointments made | 10.1 | 10.8 | 7.3 | 7.4 | -77.23 | <0.01 |
| Mean number of appointments missed | 2.4 | 5.1 | 1.1 | 2.6 | -28.24 | <0.01 |
| Missed appointment category | | | | | | |
| Zero | 548 | 42.0 | 3441 | 52.8 | 152.0 | <0.01 |
| Low | 330 | 25.3 | 1629 | 25.0 | | |
| Medium | 164 | 12.6 | 504 | 7.7 | | |
| High | 106 | 8.1 | 166 | 2.5 | | |
| **Adults** | ADHD (n = 1,304) | | Controls (n = 6,520) | | t/ $\chi^2$ | p-value |
| | M/N | SD/% | M/N | SD/% | | |
| Mean number of appointments made | 17.4 | 18.2 | 12.5 | 13.8 | -79.09 | <0.01 |
| Mean number of appointments missed | 4.0 | 8.3 | 2.1 | 4.7 | -36.25 | <0.01 |
| Missed appointment category | | | | | | |
| Zero | 398 | 30.5 | 2974 | 45.6 | 158.6 | < .0.01 |
| Low | 417 | 32.0 | 2028 | 31.1 | | |
| Medium | 278 | 21.3 | 992 | 15.2 | | |
| High | 211 | 16.2 | 526 | 8.1 | | |

M: Mean; SD = standard deviation; zero = no missed appointments, low = <1 missed appointment per year, medium = 1–2 missed appointment per year, high = >2 missed appointments per year among ADHD cases and controls

individuals with ADHD who do not receive support or treatment. This is likely to be the case in Scotland, which has few public sector services providing specific treatment to adults with ADHD.

We observed that adults, compared to young people, are more likely to miss appointments irrespective of ADHD, but also that ADHD was associated with an increased number of missed appointments even after correction for the higher number of scheduled appointments. In the sample <18 years of age, 20.7% of those with ADHD compared to 10.2% of those without had at least one missed appointment annually. The corresponding proportions in the adult sample were 37.5% vs. 23.3%.

This study did not assess whether missing appointments was associated with negative outcomes, as documented in one of our previous studies [2]. However, since patients with ADHD are more likely to miss appointments, this could be associated with some of the excess morbidity and mortality observed in individuals with ADHD in other studies [25, 26]. Due to the absence of previous studies on ADHD and the association to missed appointments in general practice we are unable to compare our findings against similar studies conducted elsewhere. In the context of specialist adult mental health outpatient services, a Danish clinic reported that 42% of their patients had ≥3 missed appointments, and a mean number of missed appointments of 2.5 (SD 2.5), with numbers increasing with time [17]. The authors identified that job instability, lower educational achievement, and poor school attendance was associated with missed appointments. Data from a Dutch forensic psychiatric outpatient service also demonstrated problems related to adults with ADHD missing appointments, as only 14.4% of adults with ADHD attending the service kept all their appointments, and out of a mean of 37.9 (SD 27.3) scheduled appointments, 6.5 (SD 6.0) were missed [18]. Missing the first or later appointments predicted a higher risk of later non-attendance [18]. Adults attending specialist or forensic psychiatric services are a vulnerable and often complex subgroup of patients with

ADHD, which could bias findings upwards. However, our findings, combined with the findings from others, does suggest that missing appointments may be a problem for individuals with ADHD in any part of the healthcare system.

Children and adolescents with ADHD were in relative terms less likely than adults with ADHD to miss appointments. While a high level of heritability of ADHD [27] could lead to parental genetic factors playing a part in non-attendance among young people, we observe the opposite pattern here. Telephone interviews with parents who had missed multiple appointments have indicated that barriers related to showing up with children included work schedules, a child's school and afterschool activities as well as high levels of stress in parents themselves [16]. Our results suggest that despite these challenges, parents or guardians who bring children to appointments may be increasingly well versed at navigating these barriers. However, the exact reasons for these differences cannot be elucidated from this data.

## Strengths and limitations

We were able to access general practice data for a very large population sample, including information about appointment attendance and morbidity. Almost all the Scottish population is registered with a GP and our sample is broadly representative of the population as a whole. GP records in Scotland are comprehensive health records and contain correspondence from all specialist services. GPs generally prescribe on behalf of NHS specialist services, but it is likely that some ADHD specialists would issue psychostimulant prescriptions until therapeutic doses are reached. We did not link secondary care mental health administrative datasets to the GP data in this study, and it is thus possible that some patients attending specialist services did not have diagnostic or treatment information transferred to their GP record. Due to the low prevalence of recorded ADHD in our dataset it is relevant to consider how this may impact findings related to missingness. It is likely that only the most severe cases of ADHD have been recognized and recorded which could lead to overestimation of missingness, but this cannot be elucidated based on the available data. However, even if we here overestimate the risk of missingness, the findings still provide important information on the potential at-risk status of these more severe cases in general practice. A further limitation is that our analyses were based only on those patients who had scheduled at least one GP appointment, thus it is important to remember that findings may only generalise to those who tend to attend general practice. Finally, we recognise that many children and adolescents would have attended with a parent or guardian. In other words, they were 'brought' to an appointment. This additional support alone may explain why children and adolescents are less likely to miss appointments in comparison to adults (irrespective of an ADHD diagnosis). However, whether a patient was 'brought' to an appointment was not coded in the record.

## Clinical implications

The low prevalence of recorded ADHD, along with the established long-term benefits of treatment, suggests that improved recognition of the condition, particularly among adults, better data recording in general practice and improved access to diagnostic and treatment services, could lead to substantial population health benefit. Our results suggest that non-attendance could potentially play a role in both delayed diagnosis and treatment. Diagnosis of ADHD is often delayed or missed, and this may impede access to effective treatments. For example, two years before a diagnosis children and young people (CYP) with ADHD attend healthcare services twice as often as CYP without. Those with a diagnosis show increased rates of physical conditions, such as asthma and eczema [28]. Early contacts that involve these early conditions may be an opportunity for earlier recognition and diagnosis of ADHD. Among those with

ADHD, whether or not formally diagnosed or treated, 'missingness' in health care should be better acknowledged and there may be potential for improved access to clinical support in general practice through a variety of sociotechnical interventions that are collaboratively developed by patients, technology developers and care providers [19].

## Conclusion

The present study was the first, by knowledge of the authors, to address the prevalence of recorded ADHD in general practice and to assess whether individuals with recorded ADHD were more likely to miss GP appointments. After adjusting for the total number of appointments scheduled we identified that for both those under and above age 18 years, those with recorded ADHD were 60–90% more likely to have missed GP appointments than the general population. Among children and adolescents with recorded ADHD, 21% missed at least one appointment annually and 8% missed two or more annually. In adults 38% missed at least one appointment annually and 16% missed two or more. These findings are of concern as missed appointments have both short and long-term consequences for the patient as well as for society as a whole. Missed appointments are not only expensive in the short term in terms of wasting financial and staff resources in an already challenged health care system. They lead to missed opportunities for providing and receiving needed care for a population at high risk of mental and physical morbidity.

## Acknowledgments

Thanks are due for the substantial contribution of Dr Christina Mohr Jensen, Aalborg University Hospital, Denmark to the drafting of this paper.

## Author Contributions

**Conceptualization:** Ross McQueenie, David A. Ellis, Andrea Williamson, Philip Wilson.

**Data curation:** Ross McQueenie, David A. Ellis.

**Formal analysis:** Ross McQueenie, David A. Ellis.

**Funding acquisition:** David A. Ellis, Andrea Williamson, Philip Wilson.

**Investigation:** Andrea Williamson.

**Project administration:** David A. Ellis, Andrea Williamson.

**Supervision:** David A. Ellis, Andrea Williamson, Philip Wilson.

**Writing – original draft:** Ross McQueenie, David A. Ellis.

**Writing – review & editing:** Ross McQueenie, David A. Ellis, Andrea Williamson, Philip Wilson.

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
