## [Decision Letter · Decision Letter 0]

5 Mar 2024

PMEN-D-24-00011

Attention-Deficit/Hyperactivity Disorder and serial missed appointments in general practice

PLOS Mental Health

Dear Dr. Ellis,

Thank you for submitting your manuscript to PLOS Mental Health. After careful consideration, we feel that it has merit but does not fully meet PLOS Mental Health’s publication criteria as it currently stands. Therefore, we invite you to submit a revised version of the manuscript that addresses the points raised during the review process.

I wish to draw your attention in particular to the reviewer comments around framing and language used throughout the manuscript, in addition to other minor revisions submitted by reviewers.

We look forward to receiving your revised manuscript.

Kind regards,

Emma Louise Giles, PhD

Academic Editor

PLOS Mental Health

Journal Requirements:

Additional Editor Comments (if provided):

Reviewers' comments:

Reviewer's Responses to Questions

**Comments to the Author**

1. Does this manuscript meet PLOS Mental Health’s publication criteria? Is the manuscript technically sound, and do the data support the conclusions? The manuscript must describe methodologically and ethically rigorous research with conclusions that are appropriately drawn based on the data presented.

Reviewer #1: Yes

Reviewer #2: Yes

Reviewer #3: Yes

Reviewer #4: Yes

2. Has the statistical analysis been performed appropriately and rigorously?

Reviewer #1: I don't know

Reviewer #2: Yes

Reviewer #3: Yes

Reviewer #4: Yes

3. Have the authors made all data underlying the findings in their manuscript fully available (please refer to the Data Availability Statement at the start of the manuscript PDF file)?

Reviewer #1: No

Reviewer #2: Yes

Reviewer #3: Yes

Reviewer #4: Yes

4. Is the manuscript presented in an intelligible fashion and written in standard English?

Reviewer #1: Yes

Reviewer #2: Yes

Reviewer #3: Yes

Reviewer #4: Yes

5. Review Comments to the Author

Reviewer #1: The authors provide a data availability statement explaining their access to the data and how this can be made available to others if required.

The manuscript focuses on a very topical and interesting issue. Offering useful insights on a challenging issue for general practice.

Some observations and queries: -

• Missed appointments were defined based on recorded time logs for appointments using methods described in previous papers. (pg8)

o Some further detail to explain the time logs would be useful here rather than the reference to other papers. It would be useful to know if missed appointments were defined any further, for example did this include cancelled appointments, appointments where children (or adults) may have been recorded as ‘not brought’ to appointments?

o Are the time logs linked to codes such as Did not attend or was not brought, as there are many potential options within systems to indicate a missed appointment.

• We observed that adults, compared to young people, are more likely to miss appointments irrespective of ADHD, (pg17)

o It may have been useful to see some discussion here about difference in child and adult attendance at appointments, recognising that children are likely to be ‘brought’ to appointments by families/parents and therefore this may account for lower missed appointments.

o This relates to my earlier point re the explanation of time logs and if this considered where children or adults were not brought to appointments where systems have recorded this data.

• Children and adolescents with ADHD were in relative terms more likely than the adults in the sample to miss appointments.

o This point appears to contradict the earlier point re adults being more likely to miss appointments.

The conclusion offers a useful account of the implications of missed appointments. Have the authors considered the system implications for missed appointments. Inflexibility in healthcare systems, appointment availability, environmental considerations when attending appointments for people with ADHD may all be compounding factors in non-attendance and are worthy of some acknowledgement as these are issues explored in other populations such as learning disabilities.

This is an interesting area of study and a very useful piece of work to support and improve access to health services and address health inequalities.

Reviewer #2: Great work! I was enjoying reading it! The work is great; you approached a very interesting and understudied field, connecting the mental health and behaviour of the patients. The study design is rigorous, and data analysis was conducted in accordance with all standards.

Reviewer #3: In my opinion, the manuscript is acceptable because it clearly and precisely explains the background of the study, the methods used, the results obtained, the conclusions, and the scope and limitations. They also include references to documents where the study procedures can be consulted in more detail.

The authors state that the results have limited clinical implications, but this work is a pioneering study on this topic.

Reviewer #4: PLOS Mental Health

Title: Attention-Deficit/Hyperactivity Disorder and serial missed appointments in general practice

Thank you for allowing me to review your manuscript, Attention-Deficit/Hyperactivity Disorder and serial missed appointments in general practice. This study examined missed appointments in General Practice (GP) and patients with ADHD. I am currently recommending a revise and resubmit, encouraging the authors to focus on shifting language to be less pathologizing (described below) and addressing some questions and limitations of the methods.

Overall:

- This paper is grounded in the pathology paradigm, which is outdated and a direct opposition to the neurodiversity paradigm, which is embraced by many disabled people. I would consider revising the paper to remove some of the pathologizing language. The content is incredibly important, and as someone who identifies as ADHD, a very relevant and personal topic to me. However, language like “exposed individuals” and “not exposed” individuals sends the message that ADHD is something bad that can be transmitted through exposure.

Introduction:

● No comments

Methods:

● Matched on sex and age

o How is sex defined? Binary option? If that is all that is available, this should be listed as a limitation of the study, as many neurodivergent people identify as gender non-conforming.

● Strong description of how sample was identified

● Description of methods in other papers

o If there is no concern with space, I would recommend that methods described in previous papers are also included in this one. As a reader, I would like to be able to read this paper and understand the strengths and limitations of the methods without needing to move to another paper to review.

Results & Discussion

● No comments

6. PLOS authors have the option to publish the peer review history of their article (what does this mean?). If published, this will include your full peer review and any attached files.

**Do you want your identity to be public for this peer review?** For information about this choice, including consent withdrawal, please see our Privacy Policy.

Reviewer #1: No

Reviewer #2: No

Reviewer #3: No

Reviewer #4: No

---

## [Editor Report · Decision Letter 1]

21 May 2024

Attention-Deficit/Hyperactivity Disorder and serial missed appointments in general practice

PMEN-D-24-00011R1

Dear Prof Ellis,

We are pleased to inform you that your manuscript 'Attention-Deficit/Hyperactivity Disorder and serial missed appointments in general practice' has been provisionally accepted for publication in PLOS Mental Health.

Executive Editor (Dr. Karli Montague-Cardoso) comments:

- Please upload the letter from West of Scotland NHS Ethics Committee and the University of Glasgow College of Medical, Veterinary & Life Sciences Ethics Committee confirming that the full study did not need health service ethics permissions as a supplementary file.

- The Editors have noted that you refer to ADHD as a mental health problem. This is not necessarily in keeping with the view of the whole community. Can you please alter the text to reflect that ADHD *can* be accompanied by changes in mental health but is not always considered to be a mental health problem by the ADHD community. Please reach out to our Executive Editor (kmontague-cardoso@plos.org) if you wish to discuss this request further.

Best regards,

Emma Louise Giles, PhD

Academic Editor

PLOS Mental Health